# Neuroinflammation, Microglia, and Cell-Association during Prion Disease

**DOI:** 10.3390/v11010065

**Published:** 2019-01-15

**Authors:** James A. Carroll, Bruce Chesebro

**Affiliations:** Laboratory of Persistent Viral Diseases, Rocky Mountain Laboratories, National Institute of Allergy and Infectious Diseases, National Institutes of Health, Hamilton, MT 59840, USA; bchesebro@niaid.nih.gov

**Keywords:** microglia, astroglia, neuron, neuroinflammation, cytokine, chemokine, PLX5622, prion, scrapie

## Abstract

Prion disorders are transmissible diseases caused by a proteinaceous infectious agent that can infect the lymphatic and nervous systems. The clinical features of prion diseases can vary, but common hallmarks in the central nervous system (CNS) are deposition of abnormally folded protease-resistant prion protein (PrPres or PrPSc), astrogliosis, microgliosis, and neurodegeneration. Numerous proinflammatory effectors expressed by astrocytes and microglia are increased in the brain during prion infection, with many of them potentially damaging to neurons when chronically upregulated. Microglia are important first responders to foreign agents and damaged cells in the CNS, but these immune-like cells also serve many essential functions in the healthy CNS. Our current understanding is that microglia are beneficial during prion infection and critical to host defense against prion disease. Studies indicate that reduction of the microglial population accelerates disease and increases PrPSc burden in the CNS. Thus, microglia are unlikely to be a foci of prion propagation in the brain. In contrast, neurons and astrocytes are known to be involved in prion replication and spread. Moreover, certain astrocytes, such as A1 reactive astrocytes, have proven neurotoxic in other neurodegenerative diseases, and thus might also influence the progression of prion-associated neurodegeneration.

## 1. Prions and Disease

Prion diseases include sporadic Creutzfeldt-Jakob disease, variant Creutzfeldt-Jakob disease, and Gerstmann-Sträussler-Scheinker syndrome in humans; bovine spongiform encephalopathy in cattle; chronic wasting disease in cervids; and scrapie in sheep and goats. As a group these diseases are often referred to as transmissible spongiform encephalopathies, and they occur naturally in humans and ruminants, but can be transmitted to rodents, nonhuman primates, felines, mustelids, and other animals.

Prion diseases are transmissible, slowly progressive, usually fatal brain diseases. Infected individuals develop vacuoles in the gray matter (spongiosis) and deposits of aggregated partially protease-resistant infectious prion protein isoforms (PrPSc or PrPres) in the brain. PrPSc is derived from the host-encoded cellular prion protein, which is sensitive to protease digestion (PrPC or PrPsen) [1]. Another hallmark of prion disease is prominent astrogliosis and microgliosis, indications shared with many neuroinflammatory and neurodegenerative disorders. The direct cause of this gliosis is unclear, but microglial and astroglial activation coincides with the detection of disease-associated PrPSc [2].

Though prion pathogenesis is not completely understood, damage and/or loss of neurons during disease is likely a major contributing factor. Neuronal damage after prion infection may occur through multiple mechanisms including excitotoxicity [3,4], inflammatory cytokine exposure [5,6,7], mitochondrial dysfunction [8,9,10,11,12], or targeted cell death through the direct interaction with the prion protein [13,14,15]. Interestingly, gliosis and PrPSc deposition precede morphological evidence of neuronal damage and neuropil vacuolation in the brain [16,17], suggesting that both PrPSc and gliosis might contribute to neuronal damage in prion disease.

Several neurodegenerative diseases including Alzheimer’s disease, Parkinson’s disease, frontotemporal dementia, and prion diseases are characterized by accumulation of aggregates of misfolded protein in the brain [18]. The particular protein or proteins involved in each of these diseases are different, but in each disease the protein misfolding appears to be spread within the brain by a seeding process where one misfolded aggregate can seed the misfolding of other normally folded molecules of the same protein by a mechanism known as “seeded polymerization” [19,20]. In the case of prion diseases, seeded amplification results in increased levels of the misfolded protein and spread to adjacent brain regions. In addition, extracts from these brains can transmit prion disease to new individuals by experimental, iatrogenic or natural routes [21]. The realization that seeded polymerization is a similar process, not only in infectious prion diseases, but also in some other non-infectious neurological diseases, has led to a resurgence of interest in studies of prion-like effects in many neurodegenerative diseases [22]. 

## 2. Neuroinflammation in Prion Disease

Originally it was assumed that prion diseases did not elicit an immune response due to the absence of a humoral response to PrPSc and a lack of interferon production in the infected host [23]. Later, it was discovered that an assortment of proinflammatory cytokines and chemokines were increased in the CNS in response to prion infection. The neuroinflammation is likely produced by the cells found within the CNS, since infiltration of leucocytes from the periphery is limited and weakly detectable only at the later stages of clinical disease [6,24,25].

Various high-throughput techniques such as microarray expression profiling [26,27,28,29,30,31,32,33,34] and quantitative bead-based suspension array systems [2,7] have elucidated transcriptional and protein changes in brains of prion-infected mice relative to controls. It is now accepted that prion diseases have a neuroinflammatory component that may play a critical role in neurodegeneration [35], with increases in numerous proinflammatory cytokines and chemokines such as IL-1α and β, IL-12p40, TNF, CCL2–CCL6, and CXCL10 in the brains of mice with clinical disease.

A more sensitive and focused approach using high-density qRT-PCR arrays has allowed us to assess the temporal changes in numerous genes comparing scrapie strain 22L-infected mice at 44, 70, 94, and 131 dpi to mock-challenged mice [5]. Several proinflammatory cytokines are increased at 44 dpi, and the number increases as prion disease advances. It appears that neuroinflammation during prion disease progressively intensifies with time, leading to chronic inflammation that probably contributes to prion pathogenesis (Figure 1).

Several of the genes/proteins found to be chronically increased during scrapie infection could potentially be damaging to the host CNS. Expression of *Oas1a*, *Isg15*, *Tnfsf11*, *Olr1*, and *Ccl5* are associated with triggering apoptosis in cells [36,37,38,39,40,41], and expression of *Cxcl10*, *Ccl2*, *A2m*, and *Tnf* can contribute to neurotoxicity in other disease models [42,43,44,45,46,47], suggesting that signaling through these proinflammatory effectors and their receptors can lead to damage. Remarkably, different strains of mouse-adapted scrapie induced similar, but not identical, profiles of increased inflammatory genes and proteins (Figure 2).

qRT-PCR array analysis of 10 signal transduction pathways revealed that the JAK-STAT and NF-κB pathways are substantially activated in prion-infected mice [5]. Over 50% of the proinflammatory genes identified as increased during prion disease could be activated by NF-κB. Furthermore, many additional genes identified are known to be regulated by specific STAT complexes. Phosphorylated STAT1 (pSTAT1) and pSTAT3 are increased when mice are infected with scrapie strain ME7 [48]. Similar to these findings, we identified an increase in total STAT1α, as well as an increase in pSTAT1α and pSTAT3, in our 22L-scrapie model [5].

Phosphorylated STAT proteins can act synergistically with NF-κB, and this might be occurring during prion infection. pSTAT3 and NF-κB have been shown to affect transcription at the promoters controlling many of the genes that are increased in the CNS during prion disease (i.e., *Cxcl10, Ccl4,* and *A2m*) [49,50,51,52,53], and together they strongly influence the expression of acute phase proteins such as haptoglobin, ceruloplasmin, α1-antichymotrypsin, and serum amyloid A [52,54], which are increased in the serum and brain during scrapie infection [5,55,56,57]. Moreover, components of the NF-κB complex, like RELA, can interact directly with STAT3 to alter transcriptional activity [58,59,60]. In addition, evidence for synergism of NF-κB and Stat1 has also been shown for the expression of many inflammatory genes such as *Ccl5*, *Cxcl9*, *Nos2*, and *Icam1* [61,62,63,64,65] that are also increased during scrapie infection. Thus, synergy might be important in neuroinflammation during prion infection of the CNS. Though several signal transduction pathways contribute to neuroinflammation in the prion-infected brain, the direct cause of pathway activation is unclear.

Several mouse models overexpressing or deficient in specific immune effectors have been assessed to understand the role of neuroinflammation during prion disease. A single deficiency in most inflammatory genes has no effect on the course of prion disease or disease pathology. Our lab intracerebrally inoculated mice lacking *IL12-p40*, *IL12-p35*, *Cx3cr1*, *IL1rn*, *C3aR1*, and *C5ar1* [2,5,66,67] with prions and saw no effect on disease. Furthermore, other labs have evaluated mice deficient in such immune genes as *Tnf* [68,69,70], *Tnfr1* [69,71], *IL-6* [68], *Ccr2* [70], *Ccr5* [70], and *Cxcr5* [72], but again the loss of expression had no effect on prion pathogenesis. The effect of deleting some genes, such as *Ccl2* [73,74] and *IL-10* [70,75], on prion disease have proven controversial by both shortening and extending survival times in mice depending on the study. Deletion of *IL1r1* prolonged the incubation time in infected mice [76], but prion infection of mice deficient in *IL-4* [75], *IL-13* [75], *Cxcr3* [77], *Tlr4* [78], and *Tlr2* [67] shortened the incubation time. Though, the deletion of several immune effectors does alter prion pathogenesis, it is important to be cognizant that the disease still progresses and is fatal. The loss of any one immune effector may be compensated by another intact or overlapping system. Thus, it is not surprising that using any single deletion mutation might yield, at best, only partial protection from prion infection. Alternative approaches such as network analysis to identify and alter “signaling bottlenecks” may be necessary to fully understand the role of neuroinflammation during prion pathogenesis.

Neuroinflammation is common in many neurodegenerative diseases including multiple sclerosis, and prion-like diseases such as Alzheimer’s disease, Parkinson’s disease, and tauopathies [79,80,81,82,83,84]. Therefore, treatment to reduce neuroinflammation may also reduce the pathology associated with prion-like diseases. Repeated injections of prednisone acetate [85] or arachis oil [86] into scrapie-infected mice inoculated intraperitoneally were effective at extending survival in some cases by more than 200 dpi, yet treatment with prednisone was ineffective with mice inoculated intracerebrally [85]. In studies using rats inoculated intracerebrally with Creutzfeldt-Jakob disease and treated with either indomethacin or dapsone [87], only dapsone treatment increased survival time. In addition, ibuprofen treatment of intracerebrally scrapie-infected mice was inconclusive due to early termination because of severe adverse side effects in the treated infected [88].

Statins have been shown to lessen inflammation in various models of neurodegenerative disease [89,90]. Atorvastatin and simvastatin affect neuroinflammation in mouse models of Parkinson’s disease by reducing proinflammatory cytokines in the brain [91,92,93,94]. Furthermore, in rodent models of Alzheimer’s disease, atorvastatin reduces the production of proinflammatory cytokines and decreases the number of microglia in the hippocampus [95,96]. Similarly, in studies using the experimental autoimmune encephalomyelitis rodent model for multiple sclerosis, statins reduce proinflammatory cytokines, increase anti-inflammatory responses, decrease infiltration of monocytes into the central nervous system, and decrease adhesion molecule expression on immune cells [97,98,99,100]. The efficacy of statin therapy in human clinical trials to reduce neurodegeneration and neuroinflammation remains controversial. Some clinical investigations report that statin therapy reduced the incidence of Parkinson’s disease [101,102,103], but others conclude that statins are ineffective in halting progression, risk, or associated dementia in Parkinson’s disease [104,105]. Clinical trials to assess the effectiveness of statins on Alzheimer’s disease progression have also produced mixed findings, with some groups reporting that statin therapy improved cognition and enhanced memory in Alzheimer’s disease patients [89,106,107,108], but others reporting no benefit from statin treatment [89,109,110,111]. Likewise, the findings from clinical trials with multiple sclerosis patients [112,113,114,115,116] has led investigators to conclude that statin treatment may offer little benefit.

Statin treatment in mouse-adapted scrapie models using simvastatin (Zocor) [117,118,119] or pravastatin (Pravachol) [120] also describe modest statistically significant improvements in survival times. We investigated the ability of two Type 1 statins (simvastatin and pravastatin) and the Type 2 lipophilic statin atorvastatin (Lipitor) to reduce neuroinflammation and improve survival during prion infection using a blinded protocol [121]. Gliosis and PrPSc deposition in the CNS were similar in statin-treated and untreated infected mice. Furthermore, the time to euthanasia due to advanced clinical signs was not changed in any of the groups of statin-treated mice relative to untreated mice [121], a finding at odds with previous reports. Ultimately, these studies indicated that none of the three statins tested was effective in reducing scrapie-induced neuroinflammation or neuropathogenesis. Based on the summation of the available data, statin therapy is unlikely to benefit individuals with prion disease.

## 3. In Vivo Assessment of Microglia in Prion Disease

Microglia are important first responders to foreign agents and damaged cells, but also have many essential functions in the healthy CNS, including neurodevelopment, synapse sensing and remodeling, and maintaining homeostasis through surveillance and phagocytosis within the brain parenchyma [122,123,124,125]. Though microglia are important in defense and maintenance of the CNS, there is evidence that their activation can lead to a dysfunctional microglial phenotype that can contribute to or exacerbate many neurological diseases including Alzheimer’s disease [126,127], multiple sclerosis [128], Parkinson’s disease [129], and HIV-associated dementia [130]. In experimental models of prion infection, microgliosis occurs prior to neuronal loss and spongiform change in the brain [131,132], and much of the inflammatory response associated with prion disease is attributed to the activation of microglia [133].

Microglia are derived early during embryogenesis from erythro-myeloid progenitors in the yolk sac that migrate to colonize the CNS rudiment [134,135]. Once in the CNS, these cells become self-renewing [136], but they are dependent on survival for continual signaling through CSF-1R, a tyrosine kinase receptor [137,138,139]. CSF-1R has two known ligands, CSF-1 and IL-34, which are produced and secreted predominately by astrocytes and neurons in the CNS [140]. In mice deficient in IL-34, microglia are reduced by 17-64% of normal in 5 brain regions [141]. Similarly, in mutant op/op mice, which are unable to produce CSF-1, microglia are reduced by 34-47% of normal in the brain [142]. Thus, manipulation of CSF-1R function might be an effective way to alter microglial function during scrapie infection.

Initial studies using the CSF-1R inhibitor GW2580 found that treatment of prion-infected mice between 98 and 126 dpi decreased microglia proliferation and lead to an increase in survival of infected mice by 26 days [143]. In addition, there was a delay in various behavior-associated clinical signs of disease and in neurodegenerative pathology with GW2580 therapy. These beneficial effects correlate with a 50% reduction in microglia in both the hippocampus and thalamus in clinical mice, a decrease in expression of genes associated with the M1 phenotype, and an increase in genes associated with the “M2” phenotype. The authors speculate that this switch in microglial gene expression profile might increase survival time by reducing the prion-induced neurotoxic effect of microglia. Contrary to these findings, we and others have demonstrated that reduction or depletion of microglia is detrimental during prion disease.

As mentioned earlier, IL-34^−/−^ mice have reduced numbers of microglia in the brain [141]. In scrapie-infected IL-34^−/−^ mice, a significant decrease in survival of 14–21 days was observed; however, the mice did not show evidence for depletion of microglia during prion infection [144]. Analysis of microglia at presymptomatic and clinical time-points showed no difference between prion-infected wild-type and IL-34^−/−^ mice, which was contrary to reports by others using uninfected mice [141]. The authors concluded that IL-34 was not required for microglial activation in the presence of prion infection [144]. However, this explanation leaves open the question of what caused the decrease in scrapie survival in the IL-34^−/−^ mice that had normal levels of activated microglia. One possibility is that in the absence of IL-34 a necessary function of microglia, like phagocytosis or catabolism of PrPSc, might be impaired.

To better address the contribution of microglia and the accompanying microgliosis during prion infection, we chemically ablated microglia (Figure 3) from mice using the CSF-1R tyrosine kinase inhibitor PLX5622 [145]. Depletion of microglia in the CNS 14 days post-prion infection significantly accelerated prion disease progression, astrogliosis, and spongiform change with three different scrapie strains (Figure 4).

Prion-infected PLX5622 treated mice had to be euthanized due to advanced clinical signs of disease between 20 to 32 days earlier than infected control mice depending on the prion strain. Furthermore, PLX5622-treated prion-infected mice accumulated significantly higher levels of PrPres relative to untreated mice at 80 dpi, 100 dpi, and experimental endpoint. These results indicate that microglia are important in controlling the infection and are beneficial.

Microglia exist as multiple subpopulations in the CNS that result from regional influences [146,147,148]. With the multitude of proposed activation states [149,150,151], it is possible that these subpopulations of microglia may demonstrate neuroprotective characteristics, neurotoxic properties, or both at different times during prion and prion-like neurodegenerative diseases. To address this possibility, we chemically ablated microglia from the CNS of mice that had been infected with prions for 80 days prior to administration of PLX5622 [145]. To our surprise, treated mice had a more rapid disease progression and had to be euthanized due to advanced disease approximately 33 days earlier than the untreated control group. These results were strikingly comparable to what we observed when mice were treated with PLX5622 after 14 dpi, suggesting that microglia depletion even at the later stages of preclinical prion disease also accelerates prion pathogenesis. Thus, microglia are beneficial throughout prion disease and may be most effective in the later stages.

## 4. Cell-Association Studies

Several studies indicate that scrapie, a natural prion disease of sheep and goats, consists of distinct strains that differ in incubation period, pathology, and clinical characteristics that are highly reproducible when introduced into mice [152,153,154,155,156,157]. The molecular explanation for the maintenance of diverse strain phenotypes in a single mouse strain with only one type of PrP protein sequence is not clear. However, the secondary structure of the PrPSc aggregates is known to vary among certain strains, and such structures appear to be maintained during templated replication of prions using a single primary PrP protein sequence [158]. Strain-specific differences in the regional patterns of prion-induced vacuolar neuropathology or prion deposition have also been documented in other hosts such as goats, sheep, and hamsters [152,159,160,161,162]. Additional studies show that distinct prion strains associate with different cell types within the mouse brain.

Immunohistochemical analysis of brains following microinjection of scrapie strains 22L, RML, or ME7 prions into the striatum, indicate prion-cell associations are strain specific [6]. 22L prions accumulate around parenchymal astroglia in all areas distant from the needle track including lateral cortex, thalamus, hypothalamus, and substantia nigra as early as 20 to 40 dpi. In contrast, strain ME7 PrPSc rarely localizes with astroglia, microglia, or oligodendroglia, but instead associates primarily with neurons and neuropil after 60 dpi [6]. This is similar to studies using mice at clinical times [163,164]. Figure 5 shows a comparison of PrPSc cell-association between 22L and ME7 in various brain regions of infected mice. Interestingly, strain RML exhibits a mix of the properties seen with 22L and ME7 infections. In the thalamus and cortex, RML prions colocalize mostly with astroglia, akin to 22L. However, in substantia nigra and hypothalamus, RML prions colocalize not only with astroglia, but with neurons and neuropil, like that of ME7 [6]. These findings with strain RML are analogous to studies using the closely related scrapie strains 79A and 79V, where prions are associated with neurons and astroglia in several brain regions at clinical times [163].

Astrocytes and neurons in healthy mice or human brains express similar amounts of *Prnp* transcript (Figure 6) [165,166], and targeted expression of *Prnp* in neurons or astrocytes alone is adequate to convey susceptibility to prion infection in mice [167,168,169]. Even though disease in astrocyte specific-expressing mice is much slower, likely due to lower than normal gene transcription, these mice still present with neurodegeneration and gliosis. Though the mechanism is unknown and the influence of prion-astrocyte cell association of strains 22L, RML, 79A, and 79V is unclear, new studies have revealed a subset of reactive astrocytes that are neurotoxic in several neurodegenerative diseases [170,171]. We speculate that this could be true in prion disease as well. Additionally, prion strains 22L and RML seem to progress more rapidly than strains like ME7 that associate primarily with neurons [6,152], thus it is possible that prion association with astrocytes results in higher local levels of prion synthesis. Greater prion production could lead to hastening the disease tempo. Furthermore, astrocytic PrPSc has been previously shown to mediate neuronal damage indirectly by interaction with adjacent neuronal processes, even in the absence of PrPC expression on neurons [172]. Though the timing of proinflammatory gene upregulation is slightly different among 22L, RML, and ME7 infected mice, the cell-specificity of the prion strains did not affect the overall proinflammatory response in the brain. One could conclude that the similar patterns of neuroinflammation seen with all three scrapie strains likely share a common source, possibly neuronal damage induced directly or indirectly by PrPSc.

The selective mechanism of cell-association by specific prion strains is also not clear. Perhaps cell-specific molecules capable of acting as cofactors for strain-specific PrPSc conversion/amplification might be an explanation for these findings [173,174]. Such molecules located on the external surface of the plasma membrane of specific cell types could potentiate PrPSc localization and new generation around neurons or astroglia. Similarly, there might be intracellular factors capable of favoring intracellular PrPSc formation in specific cell types [175]. Neuropil PrPSc accumulation might be favored by factors on axons or dendrites, or on glial cell processes located in these areas. If such factors could be identified in the future, this might provide fertile ground for the development of new therapeutic drug targets against specific strains of prion diseases. This same principle might also apply to other more prevalent neurodegenerative diseases where protein aggregation within or near specific cell types is a common feature.

## 5. Conclusions

Neuroinflammation is a feature of many neurodegenerative conditions with positive and negative consequences. In prion disease, microglia have been reported to predominantly contribute to the neuroinflammatory process [133]. Microglia exist as several subpopulations, and it is plausible that microglia are multifaceted, exhibiting both neuroprotective and neurotoxic properties during the disease process. Microgliosis occurs prior to neuronal loss and spongiform change in the brain during prion disease, and there is a close association of increased microgliosis in regions with greater spongiosis and astrogliosis. Our studies using microglial ablation by PLX5622 showed that the reduction of microglial numbers in the CNS accelerates the disease [145]. In contrast, use of GW2580 to block microglial proliferation and shift the population to a more anti-inflammatory “M2” phenotype delayed the disease [143]. Ultimately, the presence of microglia is beneficial to the host, but one should not dismiss the possibility that microglial subpopulations might occur that are either directly or indirectly contributing to the neurotoxicity associated with the later stages of prion infection. Though microglia phagocytize prion protein during early disease, it appears that the removal of infectious prions becomes dysfunctional during the clinical phase of disease [176,177,178]. A strategy to combat prion and prion-like diseases may include seeking therapies that reprogram microglial responses away from proinflammatory responses and towards increasing clearance mechanisms.

Microglia are an unlikely source of PrPSc propagation because of the nearly undetectable levels of *Prnp* expression in this cell population (Figure 6) [165]. Moreover, it is clear that a reduction in the microglial cell population increases the deposition of PrPSc. Thus, microglia are not required for PrPSc deposition or prion disease. However, the factors that drive disease, neurodegeneration, and host death remain unknown. *Prnp* expression in astrocytes or neurons is sufficient to facilitate the disease [167,168], demonstrating a clear role for these cells in the disease process. Furthermore, cell association [6,16,164,179] and cell culture studies [180,181,182] also indicate that astrocytes and neurons are strong candidates for foci of PrPSc propagation and spread within the CNS. This has led to the hypothesis that a subtype of astrocyte, and not microglia, might be involved in neuronal death to a greater extent than previously thought. Neurotoxic astrocytes have recently been described and are being considered as potential contributors to the death of neurons and oligodendrocytes in several neurodegenerative disorders [170,171]. These neurotoxic reactive astrocytes, termed A1 astrocytes, might also be responsible for the neurodegeneration associated with prion infection.

## Figures and Tables

**Figure 1 viruses-11-00065-f001:**
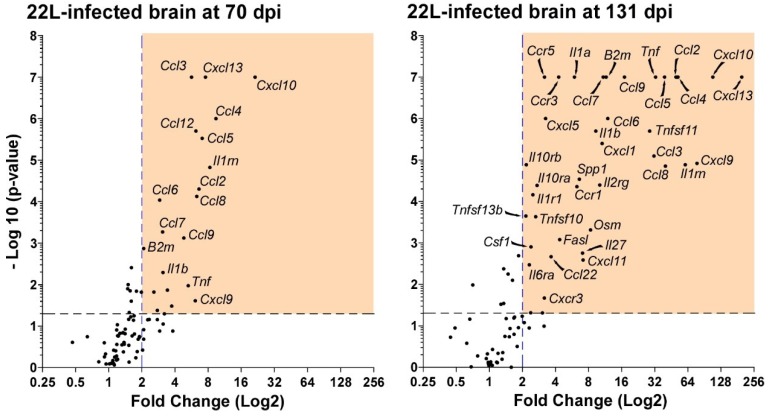
Quantitative reverse transcription-polymerase chain reaction (qRT-PCR) analysis of expression of 84 proinflammatory genes in the brains of mice at 70 compared to 131 days after prion infection. Each black dot correlates to the average gene expression seen in a minimum of three mice. The orange box is the region containing proinflammatory genes that are statistically significant by *t*-test (*p* ≤ 0.05, the black dashed line) and greater than 2-fold increased (the blue dashed line) relative to mock infected control mice. The number and magnitude of the upregulated proinflammatory genes in the brain of prion infected mice intensifies as a function of time.

**Figure 2 viruses-11-00065-f002:**
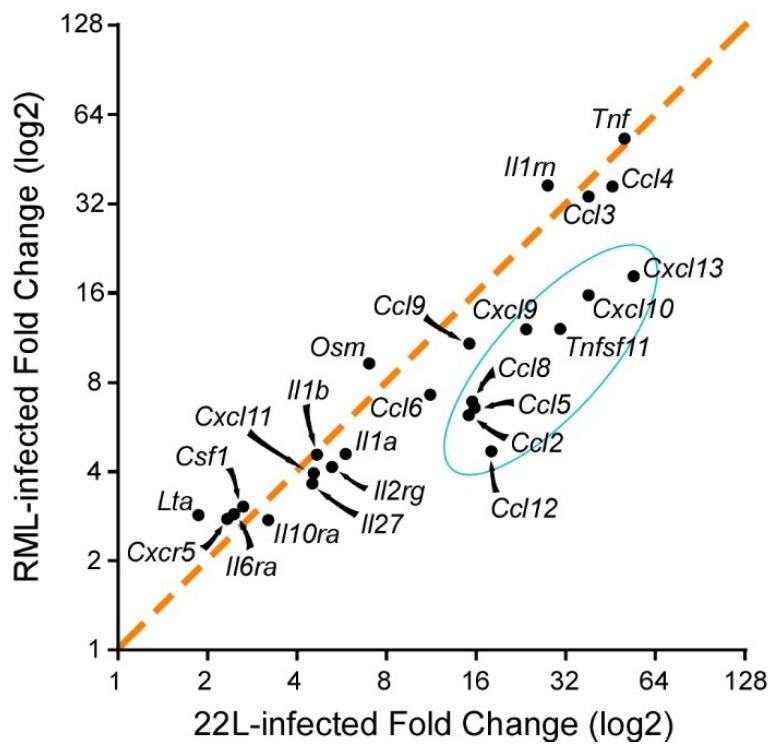
Comparison of the top 25 proinflammatory genes that are upregulated in the brain of clinical mice when infected with prion strains RML and 22L. The orange dashed line represents a one-to-one correlation in fold change of gene expression. Most of the changes are similar, thus close to the orange dotted line. There is a cluster of eight proinflammatory genes, blue circle, that are more highly altered with 22L infection, but the overall inflammation is comparable during the clinical phase of the disease regardless of prion strain.

**Figure 3 viruses-11-00065-f003:**
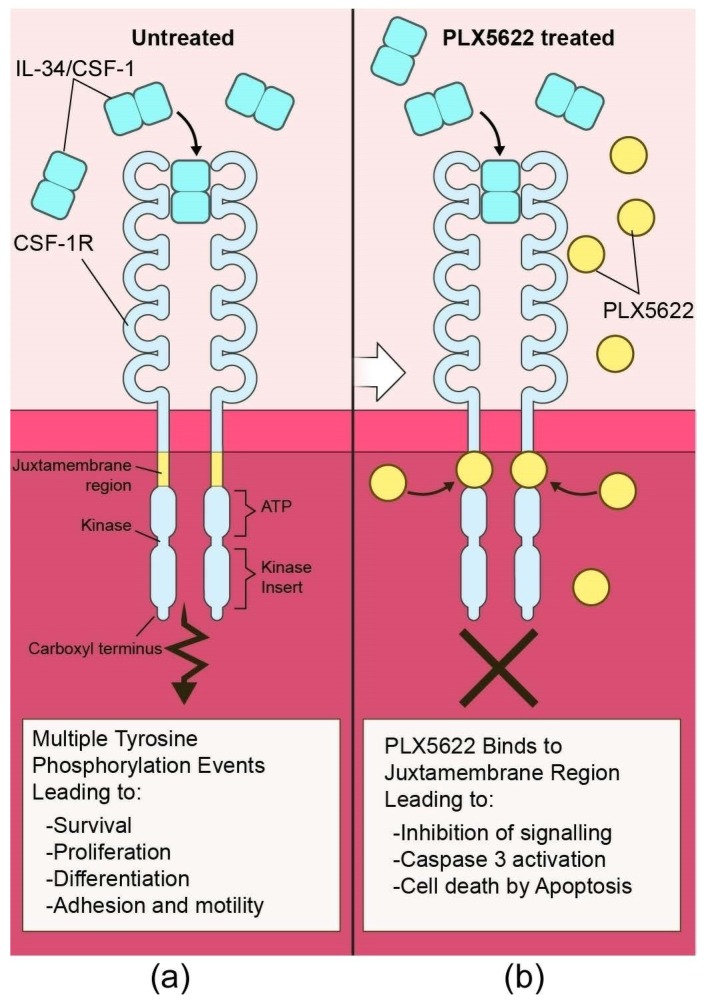
**Representation of the CSF-1 Receptor (CSF-1R) interaction with the tyrosine kinase inhibitor PLX5622.** Under normal conditions (**a**) CSF-1R interacts with either CSF-1 or IL-34 to promote phosphorylation at a minimum of eight tyrosine residues within the cytoplasmic domains. These phosphorylation sites serve as docking points for many proteins and lead to downstream signaling events. Inhibition of CSF-1R with PLX5622 (**b**) causes a cessation of signaling through this receptor in microglia, which is critical for their survival and proliferation. Microglia are eliminated from the CNS by activation of Caspase 3, leading to death by apoptosis.

**Figure 4 viruses-11-00065-f004:**
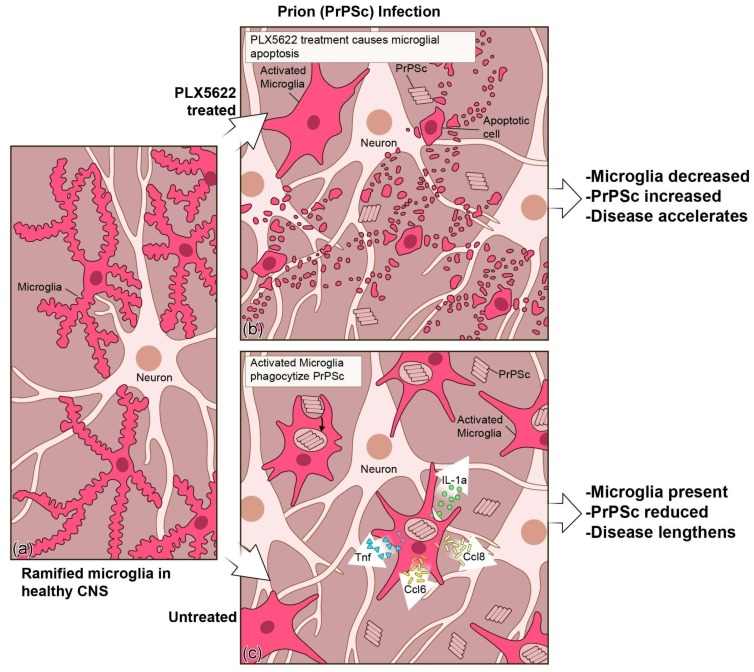
**Illustration of prion disease with and without PLX5622 treatment to reduce microglia in the Central Nervous System (CNS)**. (**a**) Microglia in the healthy CNS typically have a ramified appearance as they surveil their environment. (**b**) When prion-infected mice are treated with PLX5622, microglia are reduced, the level of PrPSc increases, and the disease process is accelerated. (**c**) When prion-infected mice are untreated, more microglia are present to phagocytize PrPSc and to produce microglial specific proinflammatory effectors like TNF, CCL6, CCL8, and IL-1a. The presence of microglia reduces the PrPSc burden and lengthens the disease.

**Figure 5 viruses-11-00065-f005:**
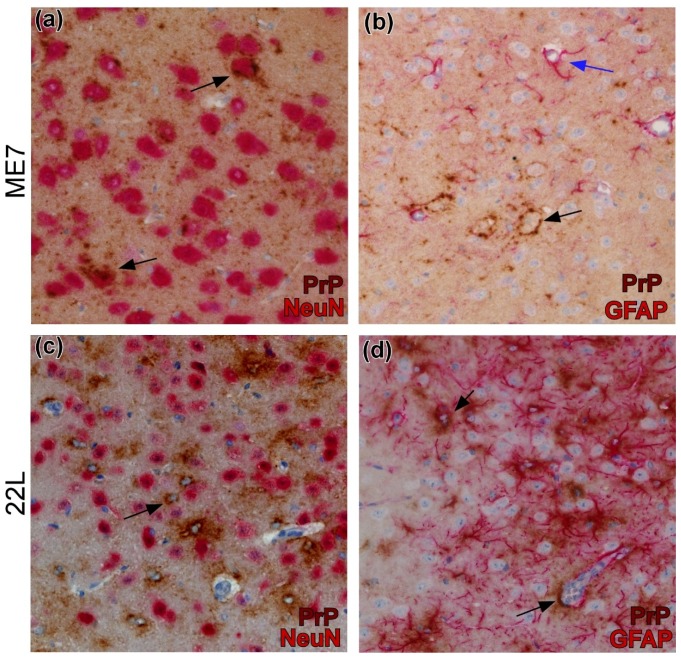
Use of dual-staining immunohistochemistry for detection of PrPSc from scrapie strains ME7 and 22L associated with neurons or astrocytes in brain. (**a**) PrPSc (brown) surrounds ME7-infected neurons (arrows) in amygdala that are detected by anti-NeuN (red). (**b**) In hypothalamus, ME7 PrPSc surrounds neuronal cell bodies (black arrow). Astrocytes detected by anti-Glial Fibrillary Acidic Protein (GFAP) (red) have no PrPSc and are not infected (blue arrow). (**c**) PrPSc (brown)-expressing 22L-infected cells (arrow) are distinct from NeuN-stained (red) neurons in thalamus. (**d**) PrPSc (brown) expressing 22L-infected cells (arrows) are associated with GFAP-stained (red) astrocytes in thalamus.

**Figure 6 viruses-11-00065-f006:**
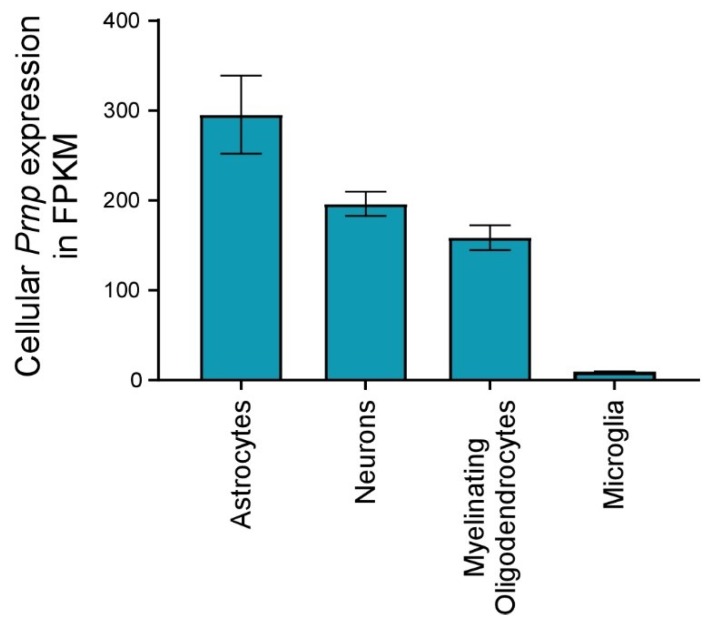
**Cellular *Prnp* expression by various brain cells in healthy mice.** RNA seq data of purified cells isolated from health mouse brain was acquired from the work of Zhang et al. and represented here in fragments per kilobase million (FPKM) for comparison [165]. The blue columns are the average of two replicates of pooled animals for each cell type and bars represent the standard deviation. Astrocytes express the most *Prnp*, while microglia express very little.

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
