# Peer review of "Neuroinflammation, Microglia, and Cell-Association during Prion Disease"

_viruses, 2019, doi:10.3390/v11010065_

Reviewer 1 Report

This is a comprehensive review on the role of inflammation in prion diseases. The message coming from this review is that microglia is neuroprotective, whereas neurons and astrocytes are "propagators" of the disease. This is similar to a recent review (Aguzzi and Zhu, J Clin Invest 2017, 127:3230).

The authors draw studies from external sources and from their lab studies to illustrate the above. In particular, they have highlighted the discrepancies regarding the usage of statins in mitigating neurodegeneration in mouse prion models.

While the authors have extensively citing animal studies and the in vivo assessment, they have also included cell-association studies.

Overall, this is a balanced review examining inflammation at different biological levels in prion diseases.

Author Response

We thank the reviewer for their comments on our submitted review article, and we are glad that they felt it was useful, informative, and balanced in its presentation of the information.

Reviewer 2 Report

In this review Carroll and Chesebro assess some of the scientific information that is available regarding neuroinflammation in prion disease and the role of proinflammatory cytokines and their contribution to prion pathogenesis. The authors attempt to assess the role of microglia in both damaged cells and also in healthy CNS tissue. They pay particular attention to various scrapie strains in murine models and compare some of their own data with that of other researchers. They look at a variety of prion diseases including the effect of chemical ablation of microglia in a mouse model and the resulting PrPres accumulation relative to untreated mice.

The authors have gathered together some interesting papers and have made a really useful review for both specialists in the field and researchers who need some guidance as to where to go to find relevant information regarding neuroinflammation, neurotoxicity and prion-associated neurodegeneration.

The role of prion pathogenesis is far from fully understood and there are many schools of thought relating to the role of microglia, neurons and astrocytes. This current review article gives a fairly unbiased view of the research that has been published to-date and also presents supporting data from their lab that highlights other suggested reasons or hypotheses that could be relevant to the field. 

Much of the already published data highlight various contradictions within data sets and between experiments or between research groups. However, there are usually subtle differences within the experimental designs that allow for those differences or at least allow some explanation as to why results do not appear to tally. I personally think that there are always numerous possibilities as to why different research groups generate different results and I think that quite often, all the results are valid but they need careful description and analysis. They often just need mindful explanation and careful collation in order to see that there are similarities and there are supporting reasons for the apparent discrepancies.

In science there are many experiments that do not appear to fit at first glance, but on further reflection, they can be shown to be supportive of each other or in the worst case they can be shown to have very different experimental designs that actually clarify the divergent results.

The prion field is still undecided regarding the role of microglia with some researchers suggesting they have little effect on prion disease progression and others stating they are immensely important in this regard.

The current authors have very eloquently summarised the current state of affairs and have given some careful and fair clarity to both sides of the argument. I think they have demonstrated that there is a fair amount of support for the view that microglia exist as several subpopulations and they are multifaceted in that they can exhibit both neuroprotective and neurotoxic properties at various stages during the disease process. With this being the case, it is completely understandable that different research laboratories generate wildly different data and that on occasions there seems to be no compatibility between studies.

As we all know, science, and especially animal studies, can have a variety of outcomes depending on prion strain, inoculation route, dose of inocula, age of the animals in question, volume of inocula, genotype of the inoculated animal and so on. It is therefore extremely useful when a review of this nature gives a clear and fair summary of numerous studies and offers up suggestions as to why there are differences between data and research groups.

I found the review very informative with good author comments, comparisons and hypotheses. They have given a clear and articulate description of previously published work from themselves and other authors.  Personally I was particularly interested in the 22L, RML, ME7 scrapie strain comparisons as these show some nice strain specific differences.

The authors make some good general comments regarding the use of statins and the fact that therapy with these is very unlikely to provide any benefit to prion diseased individuals. The suggestion that microglia depletion at later preclinical prion disease also accelerates prion disease pathogenesis is extremely relevant for prion researchers. The point that microglia may be most effective at the later stages of disease progression is an exciting proposition and will hopefully be proven to be useful in time. 

This review is a useful tool for many scientists and gives guidance as to where to look for relevant information. It will be beneficial for young scientists as it presents a balanced overview of a very complicated area of research and a scientific field that can often cause arguments as to who is right and who is wrong. There are always many grey areas but this goes some way to making it clearer in a readable manner and with some careful attention to detail.

Major Comments:

1. This is a very nicely written, clearly explained Review

2. The Figures are clearly presented and demonstrate the points being made in a concise manner

Minor points:

3. There are a few typographical errors

such as line 86 - …..greater than2-fold…

Line 148 …..severe….

Figure 4 (PrpSc or PrPSc??)

Figure 4 legend (lines 235 and 236)

Author Response

We thank the reviewer for their comments on our submitted review article, and we are glad that they felt it was useful, informative, and balanced in its presentation of the information.

We have corrected the typographical errors noticed by the reviewer and we thank them for their keen eye.  The changes are noted in the revised manuscript using the track changes function. We have also corrected Figure 4 as requested by replacing “PrpSC” with “PrPSc”.